# Association between the Sociodemographic Characteristics of Parents with Health-Related and Lifestyle Markers of Children in Three Different Spanish-Speaking Countries: An Inter-Continental Study at OECD Country Level

**DOI:** 10.3390/nu13082672

**Published:** 2021-07-31

**Authors:** Cristian Álvarez, Iris Paola Guzmán-Guzmán, Pedro Ángel Latorre-Román, Juan Párraga-Montilla, Constanza Palomino-Devia, Felipe Augusto Reyes-Oyola, Lorena Paredes-Arévalo, Marlys Leal-Oyarzún, Isabel Obando-Calderón, Mauricio Cresp-Barria, Claudia Machuca-Barria, Sebastián Peña-Troncoso, Daniel Jerez-Mayorga, Pedro Delgado-Floody

**Affiliations:** 1Quality of Life and Wellness Research Group API4, Department of Health, Universidad de Los Lagos, Osorno 5290000, Chile; cristian.alvarez@ulagos.cl; 2Faculty of Chemical-Biological Sciences, Universidad Autónoma de Guerrero, Chilpancingo de los Bravo 39087, Mexico; pao_nkiller@yahoo.com.mx; 3Department of Didactics of Corporal Expression, University of Jaen, 27301 Jaen, Spain; platorre@ujaen.es (P.Á.L.-R.); jparraga@ujaen.es (J.P.-M.); 4Faculty of Education Sciences, Universidad de Tolima, Ibagué 730006299, Colombia; cpalominod@ut.edu.co (C.P.-D.); fareyeso@ut.edu.co (F.A.R.-O.); 5Department of Health, Universidad de Los Lagos, Osorno 5290000, Chile; lorena.paredes@ulagos.cl (L.P.-A.); marlys.leal@ulagos.cl (M.L.-O.); isabel.obando@ulagos.cl (I.O.-C.); 6Faculty of Education, Universidad Católica de Temuco, Temuco 4780000, Chile; mauriciocrespbarria@gmail.com; 7Faculty of Health Science, Universidad Católica de Temuco, Temuco 4780000, Chile; cmachuca@uct.cl; 8Institute of Educational Sciences, Universidad Austral de Chile, Valdivia 5091000, Chile; sebastian.pena@uach.cl; 9Faculty of Education and Culture, Universidad SEK, Santiago 5110566, Chile; 10Faculty of Rehabilitation Sciences, Universidad Andres Bello, Santiago 7591538, Chile; daniel.jerez@unab.cl; 11Department of Physical Education, Sport and Recreation, Universidad de La Frontera, Temuco 4780000, Chile

**Keywords:** physical fitness, children, nutritional level, physical activity

## Abstract

The purpose of this cross-cultural study was to determine the association between the sociodemographic background of a child’s parents (i.e., their socioeconomic level, marital status, and educational level) with the child’s lifestyle (i.e., Mediterranean diet (MD), physical activity (PA) and screen time (ST)), and health markers. Material: This cross-sectional study included 1273 children, from Chile (*n* = 496), Colombia (*n* = 340), and Spain (*n* = 437). The sociodemographic information together with the lifestyle and health markers of the children were measured. There was an inverse association between a low or medium-low socioeconomic level for the parents of Chilean children and handgrip strength (β −0.61, *p* < 0.001); meanwhile, for Spanish children, an inverse association between a low or medium-low socioeconomic level and PA after school (β −0.58, *p* = 0.016), lifestyle (β −0.74, *p* = 0.015), and with MD adherence (β −0.86, *p* = 0.004) was found. The risk (i.e., by odd ratios (OR)) of being divorced/separated parents marital status showed an inverse association with abdominal obesity (OR 0.21, *p* = 0.045) in Spanish children; however, the parent’s marital status and a low educational level were risk factors for the suffering of a low nutritional level in Colombian children (OR 2.02, *p* = 0.048; OR 2.49, *p* < 0.001, respectively). On the other hand, a low educational level for parents reported for Chilean children had a positive association with ST of ≥4 h per day (OR 1.82, *p* = 0.020). In conclusion, in Spanish-speaking children, the lifestyle and health markers of the children are affected by the sociodemographic background of their parents; however, these effects could be moderated by the socio-cultural and economic status of their countries as members of the OCDE; therefore, it is essential to develop policies that decrease these gaps, so that children who are under-resourced can reach their full potential.

## 1. Introduction

Childhood is a relevant stage of life during which many physiological and psychological changes that are crucial for the later stages of life take place [1]. Unfortunately, the lifestyle of children today is affected when their parents have a low socioeconomic level [2]. Likewise, it is important to mention that children from schools with a low socioeconomic level had reported more barriers and fewer facilitators to obtain a healthy lifestyle with good levels of physical activity (PA) than their counterparts from schools with a high socioeconomic level [3]. A previous study has indicated that health is correlated with people’s socioeconomic status and lifestyle [4]. Concerning this, it has been reported that the socio-economic development of the country, measured as per capita gross domestic product, is the variable most strongly associated with better health in different European countries (after controlling for individual-level characteristics) [5]. For example, Olaya et al. [6] indicated that better macro-economic indicators are related to lower numbers of children who are overweight. Additionally, the current evidence mentions that parent**s**’ sociodemographic factors have pronounced effects on the prevalence of excess weight and obesity in children [7]. Thus, a recent study reported that socioeconomic status was strongly associated with childhood obesity early at the elementary school, where Hispanic children showed a 60% to acquire obesity than white American peers [8]. On the other hand, in Latin America, important negative changes have also been described by other socio-cultural factors (i.e., lower educational and socioeconomic levels from parents), and these produce a nutritional transition process [9].

Negative changes in lifestyle such as a decrease in PA and an increase in screen time (ST) in children affect their healthy psychological, physical, and social development [10]. The PA, for example, has reported an association with well-being in children; instead, ST has been linked negatively with psychological and physical health [11], and several cardiometabolic risk factors (CRF) [12]. Moreover, active lifestyle with high PA levels and low ST in children has commonly been studied in the health context; however, it is reasonable to expect that its promotion also involves enhancing emotional stability, cognitive skills, psychological well-being, learning process, and in turn, indirectly, academic achievements [2,13], therefore, the study of factors related to PA and ST in childhood has garnered increasing attention by the scientific community.

On the other hand, there is evidence that adhering to a Mediterranean diet (MD) is associated with positive effects on health [14], and MD has also been presented as part of the solution for CRF [15]. In this sense, it has been observed that the MD is rich in plant-based foods and low in animal foods, which are both healthier and exert a lower impact on the environment [16]. Nowadays, for example, the new environmental dimension according to the food production represents a major cause of ecological pressure on the natural environment, and diet links worldwide human health with environmental sustainability [17]; therefore, it is important that countries of different continents take into consideration the environment and the economy to develop MD dietary guidelines [18,19].

Looking at the health markers for children, childhood obesity at the preschool stage is a global phenomenon in developed and undeveloped countries, where the nutritional transition (i.e., the change from underweight to an overweight disease due to the industrialization processes) can explain the coexistence of obesity and nutritional deficiencies; moreover, positive energy balance behavior is associated with obesity [20]. Likewise, abdominal obesity (AO) is a major clinical and public health issue compared with generalized obesity in children, and due to AO, representing central obesity, this is more strongly correlated with CRF [21] and seems to have increased during the last few decades [22]. Moreover, it has been reported that AO in children increases CRF such as dyslipidemia, hypertension, and hyperglycemia [21]. Another negative effect of a bad lifestyle on health markers is a decrease in physical fitness, which is considered to be one of the most important health markers across the life course [23] because it is a powerful marker of health in early childhood and later in life [24,25]. Moreover, lower fitness has been associated with higher body fatness parameters and higher blood pressure levels in schoolchildren [26].

On the other hand, the goal of the Organization for Economic Cooperation and Development (OECD) is to shape policies that foster prosperity, equality, opportunity, and well-being for all, and its members work with other peers, organizations, and stakeholders worldwide to address the pressing policy challenges of our time. Spain has a high GDP per capita, Chile has an average GDP per capita, and Colombia has a lower GDP per capita, and these three countries present different socio-cultural and economic characteristics. An evaluation of children’s lifestyles in countries with different economic and sociodemographic characteristics provides very relevant information for the development of policies that contribute to the integral development of children and gives a better understanding of the factors associated with children’s lifestyle. Therefore, the purpose of this study was to determine, through a cross-cultural study in Chile, Colombia, and Spain, the association between the sociodemographic backgrounds of parents (i.e., socioeconomic level, marital status, and educational level) with the lifestyle (i.e., MD, PA after school, and ST) and health markers of their children. 

## 2. Materials and Methods

This cross-sectional study included 1273 schoolchildren, from Chile (*n* = 496), Colombia (*n* = 340), and Spain (*n* = 437), selected by convenience. The children’s parents and guardians were informed about the study and provided signed written consent for their participation. Additionally, all children gave their written consent on the day of the assessment. The study was carried out under the recommendations for human studies of the Declaration of Helsinki and was approved by the Institutional Ethical Committee (ORDN°016-31012019 and ACT N°086).

The inclusion criteria were: (i) presenting informed consent of the parents and the consent of the participant, (ii) belonging to an educational center, and (iii) being between 4 and 6 years of age. The exclusion criteria were having a musculoskeletal disorder or any other known medical condition that might alter the participant’s health and physical activity levels. Moreover, schoolchildren with physical, sensorial, or intellectual disabilities were excluded. The study design is shown in Figure 1.

### 2.1. Anthropometric Assessments

The body mass index (BMI), calculated as the body mass divided by the square of the height in meters (kg/m^2^), was used to estimate the degree of obesity. The body mass (kg) was measured using a TANITA^®^ scale, the Scale Plus UM—028 model (Tokyo, Japan). The children were weighed in their underclothes without shoes, and their height (m) was estimated with a SECA^®^ stadiometer, model 214 (Hamburg, Germany) that was graduated in mm.

### 2.2. Abdominal Obesity

Waist circumference (WC) was measured by a SECA^®^ 201 tape (Hamburg, Germany) in the anatomical position of the umbilical scar [27]. The waist-to-height ratio (WtHR) was calculated by dividing WC by height. Then, to categorize AO, a greater WtHR ≥ 0.50 was used [28,29,30]. 

### 2.3. Mediterranean Diet Adherence and Physical Activity

The parents or guardians of the children filled out the information about their children. First, they completed the information about the food habits and PA patterns for their children. (1) MD adherence was assessed by the Krece Plus test [31], a tool to determine eating patterns and the relationship with nutritional status based on the MD. The questionnaire has 15 items, and the format assesses a set of items about the food consumed in the diet. Each item has a score of +1 or −1, depending on whether it approximates the ideal of the MD. The total points are added, and according to the score, the nutritional status is classified as follows: (i) low nutritional level: less than or equal to 5; (ii) moderate nutritional level: from 6 to 8; and (iii) high nutritional level: greater than or equal to 9. This questionnaire has been used in Chilean schoolchildren [32]. (2) The child’s lifestyle was evaluated with the PA Krece Plus test [31]. The Krece Plus is a quick questionnaire that classifies lifestyle according to a daily average of hours spent watching television or playing video games (screen time) and PA hours after school per week. The classification is made according to the number of hours for each item. The total points are added, and the person is classified as having a good lifestyle (men: ≥9, women: ≥8), a regular lifestyle (men: 6–8, women: 5–7), or a bad lifestyle (men: ≤5, women: ≤4) according to the lifestyle score.

### 2.4. Sociodemographic Background of Parents

An ad hoc sociodemographic questionnaire was used; information such as the educational level, marital status, and socioeconomic background (based on the parents’ socioeconomic self-perception) was collected from the parents by personal interviews of a member of the school staff. This procedure was carried out into a comfortable room in each school by country. The questionnaire was added to the Krece Plus at the beginning. 

### 2.5. Physical Fitness

To measure physical fitness, we included leg strength, cardiorespiratory fitness (CRF), and handgrip strength. The lower-body explosive strength was measured using the standing long jump test (SLJ) [33]. The SLJ test has been used in preschool and schoolchildren previously [34] and their development includes jumping a distance with both feet simultaneously on a hard surface. Following this, each student stands behind a jump line with their feet a shoulder width apart; thus, the knees are bent with the arms in front of the body and in a parallel position to the ground. From here, they swing their arms, push hard, and jump as far as possible, making contact with the ground with both feet and in a vertical position. The SLJ test is applied twice, and the best result is recorded for future analyses.

The handgrip strength was applied to measure the upper body strength using a hand dynamometer (TKK^®^ 5101 Grip D; Takei, Tokyo, Japan). This test consists of holding a dynamometer in one hand and squeezing it as tightly as possible, without allowing the equipment to touch the body. The force is applied gradually and continuously for a maximum of 3–5 s [33]. This test was performed twice, and the maximum score to each hand was registered in kilograms (kg). The mean of the scores achieved in the left and right hands were used in the final analysis. A higher score of handgrip strength indicates better strength performance.

The CRF was measured by the 10 × 20 m test, applied by the spatial structure of the Léger test [35]. The 10 × 20 m test was used on child participants that were previously validated [36]. The materials to the 10 × 20 m test required included a tape measure to limit the distance of the runway (20 m), two boxes, five balloons, and a stopwatch for the time register. The test was a 20 m shuttle test, in which children had to move five balloons from box A, located in one extreme, to another box B, located at the opposite extreme of box A. Thus, the total distance covered was 200 m, timed from the signal “go” until each participant delivered the last balloon. In this line, it did not matter if the balloon was not delivered into the box. Additionally, if the balloon was dropped during the runs, the participant had to pick it up and carry on moving. All the supervisors gave instructions to the participants about the balloon having to be held with both hands. The test allowed both running and walking to children. For this test, only one attempt was allowed by the participant. The results were recorded in seconds (s) with one decimal place. The test score registered was the running time, where a longer time indicates poorer performance, but a low time indicates great performance.

### 2.6. Statistical Analysis

Statistical analyses were developed by the STATA V.13.0. software (Stata Corp, College Station, TX, USA). The normal distribution was tested by the Kolmogorov–Smirnov test. For continuous outcomes, values are shown as mean and the 5th and 95th percentiles, whereas for categorical variables, they are presented as numbers and proportions (%). The differences by sex and country factor were tested using the Kruskal–Wallis test, and the Mann–Whitney U test. The X^2^ test was applied to compare the proportions according to the lifestyle parameters of participants. The association between the sociodemographic backgrounds of the parents and the results for their children’s fitness and lifestyle were tested using the β coefficient, where linear and logistic regression with the inclusion of odds ratios (ORs) and 95% confidence intervals (CIs) was included. Values of *p* < 0.05 were considered statistically significant.

## 3. Results

### 3.1. Sociodemographic Characteristics

Table 1 shows the anthropometric characteristics, physical fitness, and nutritional status of the children according to their country. The anthropometric parameters show that Chilean children reported a higher WtHR (*p* < 0.001) than their Colombian and Spanish peers. Regarding fitness, Colombian children reported worse results in the 10 × 20 test than their Chilean and Spanish peers. Additionally, Chilean children showed a lower handgrip strength than their Colombian and Spanish peers. The Chilean children had higher ST (in hours per day) than their Colombian and Spanish peers (*p* < 0.001). The Spanish preschoolers reported lower PA after school (*p* < 0.001), and higher MD adherence than their Latin-American peers (*p* < 0.001).

### 3.2. Prevalence of Abdominal Obesity

Figure 2 shows the prevalence of AO (a, b), lifestyle category (c, d), and nutritional level (e, f) by country, and by country and sex (b, d, f). Chilean schoolchildren reported a high prevalence of AO and a high prevalence of bad lifestyles. Colombian preschoolers reported the lowest MD adherence.

### 3.3. Parents’ Sociodemographic Characteristics

Table 2 shows the parents’ sociodemographic characteristics for the study sample, according to the country. Looking at their parents’ socioeconomic background, the Chilean children had the highest proportion of parents in the low/medium socioeconomic level (41.7%), followed by the Colombian preschoolers (37.7%) and the Spanish preschoolers (11.0%) (*p* < 0.001) (Table 2).

### 3.4. Associations between Parental Sociodemographic Background and Children’s Characteristics 

Table 3 shows the association between the parental sociodemographic status and the anthropometric, fitness, and lifestyle parameters of the children according to country. For the Chilean children, there was an inverse association between a low or medium-low socioeconomic level for the parents and SJT (β −3.90, *p* = 0.043) and handgrip strength (β −0.61, *p* < 0.001). Likewise, for the Spanish children, an inverse association was found between the parents’ socioeconomic level and PA after school (β −0.58, *p* = 0.016), lifestyle (β −0.74, *p* = 0.015), and MD adherence (β −0.86, *p* = 0.004). For the Colombian preschoolers, a low educational level showed a positive association with SJT (β 5.87, *p* = 0.016) and an inverse association with MD adherence (β −0.63, *p* = 0.004), while for Spanish children, it had an inverse association with MD adherence (β 0.58, *p* = 0.003). 

### 3.5. Association of Parental Characteristics with Abdominal Obesity (AO)

Table 4 shows the association between parental characteristics and AO, low nutritional level, and bad lifestyle in the children. Divorced/separated status for the parents presented an inverse association with AO (OR 0.21, *p* = 0.045) in Spanish preschoolers. However, the parent’s marital status and low educational level were risk factors for a low nutritional level in Colombian children (OR 2.02, *p* = 0.048; OR 2.49, *p* < 0.001, respectively). The single marital status for the parents showed an inverse association with a bad lifestyle for the Spanish children (OR 0.21, *p* = 0.012). Parents having a primary/secondary educational level showed a positive association with ST of more than 4 h a day for the Chilean children (OR 1.82, *p* = 0.020).

## 4. Discussion

The purpose of this study was to determine, through a cross-cultural study, the association between the sociodemographic backgrounds of parents (i.e., their socioeconomic level, marital status, and educational level) with lifestyle (i.e., MD, PA after school, and ST) and health markers of children. The results show that: (i) there was an inverse association between the parents having a low or medium-low socioeconomic level and the fitness, PA after school, lifestyle, and MD adherence of their children; (ii) there was an inverse association between the single marital status for the parents and Spanish children having a bad lifestyle; and (iii) for the Chilean children, there was a positive association between the low educational level of the parents and ST of more than 4 h a day. Thus, a medium-low socioeconomic, a single marital status for the parents, and a low educational level of the parents are all related to sociodemographic, lifestyle, and health markers in children of three OCDE countries of different socio-cultural and economic (i.e., based on GDP per capita) characteristics.

According to the sociodemographic parameters of the parents, a low or medium-low socioeconomic level had an inverse association with fitness health markers and positive lifestyle (i.e., PA after school and MD adherence) in children. In this sense, physical inactivity (i.e., a negative lifestyle) is recognized as a determinant of children’s low fitness and health markers [37]. Unfortunately, fitness level is a potential biomarker of health from an early age; thus, improvements in physical fitness performance could be important for the health of preschoolers [38]. In this field, a study conducted for young people reported that the ‘high’ socioeconomic level group had better CRF and musculoskeletal fitness than the ‘low’ and ‘moderate’ CRF group [39]. Additionally, it has been shown that schoolchildren from a low socioeconomic level have a lower PA level and a worse nutritional status than their counterparts with a ‘high’ socioeconomic level [40]. Another study showed that a higher socioeconomic level for the family was associated with higher PA levels in children [41]. In a similar way to our results, a study conducted for Spanish preschoolers showed that the socioeconomic level was positively associated with better scores in MD adherence [42]. Another study conducted for Australian preschoolers found that subjects with higher socioeconomic levels had diets closer to healthy guidelines for most food [43]. Furthermore, the higher socioeconomic status of the family has been associated with healthier eating, PA, and control of weight practices when compared with families with lower socioeconomic status [44].

Within our results, it is very important to mention that lifestyle is decisive for future activities in life; lifestyle has been related to children’s motor skills [45], psychological well-being [11], academic performance [2], and fitness [46]. Additionally, Manios et al. [47] demonstrated that low diet quality was associated with several sociodemographic factors (i.e., unemployment status and lower maternal educational level) in Greek preschoolers. Moreover, we found that the marital status of the parents (i.e., divorced/separated) showed an inverse association with AO in Spanish preschoolers, with Colombian children showing a low nutritional level. On the other hand, another study reported that children in cohabiting-parent and single-parent families who had experienced a prior structural family change were more likely to be obese unless they were children in single-parent families who were born to married parents when compared with peers of no married parents [48]. Similarly, Merino et al. [49] showed that there were no significant differences in fatness indicators (i.e., BMI, WC, and WtHR) of children whose parents had different marital statuses. In addition, sociodemographic background, such as monthly household income, was negatively associated with BMI in Chinese preschoolers [50]. Another study identified certain factors such as maternal pre-pregnancy obesity, maternal smoking, low education level, and excess weight of parents as significant risk factors for the development of obesity in children [51].

Likewise, in the present study, low educational level (primary/secondary level) reported in the parents of Chilean children showed a positive association with ST of more than 4 h a day. A recent study reported that daily ST was significantly lower in children whose parents had higher education levels (i.e., university education) compared with peers whose parents had a secondary or primary, or no formal education [52]. Moreover, Määttä et al. [40] indicated that parents with a high education judged it to be of greater importance to limit the ST of preschool children, compared to parents with low education. Following this, Downing et al. [53] reported that parental self-efficacy (i.e., limiting ST and imposing ST rules) was inversely associated with ST in Australian preschoolers. This suggests that better-educated parents may be more aware of the importance of PA and may provide healthier options to their children by regulating ST which is an important marker of sedentary, and obesity.

In addition, considering that socio-cultural gaps can be a permanent barrier of health status among children of different countries members of the OECD group, the reduction in the economic gaps promoted at school from the high to those children of low economic status appears as one of the most practical strategies among countries of the OECD members.

The limitations of the present study include those inherent to its transversal character. Another limitation is the reporting by parents of their children’s PA and food habits, which could mean that these data are underestimated or overestimated. We suggest that there is a need to investigate possible longitudinal effects to clarify the direction of these associations and to carry out more interventions in children’s lifestyles among OECD countries of different socio-cultural, and economic statuses. The strengths of this study are that we examined several variables that affect children’s development and contributed to a better understanding of the severe problem of parents’ sociodemographic background. Additionally, we included three members of the OECD that allow future comparisons with other studies involving schoolchildren (preschoolers and scholars) and health markers. 

## 5. Conclusions

In conclusion, in Spanish-speaking children, the lifestyle and health markers of the children are affected by the sociodemographic background of their parents; however, these effects could be moderated by the socio-cultural and economic status of their countries as members of the OECD. Therefore, it is essential to develop and promote policies that decrease the sociodemographic gaps that affect children’s lifestyle and health markers, so that children who are under-resourced can reach their full potential.

## Figures and Tables

**Figure 1 nutrients-13-02672-f001:**
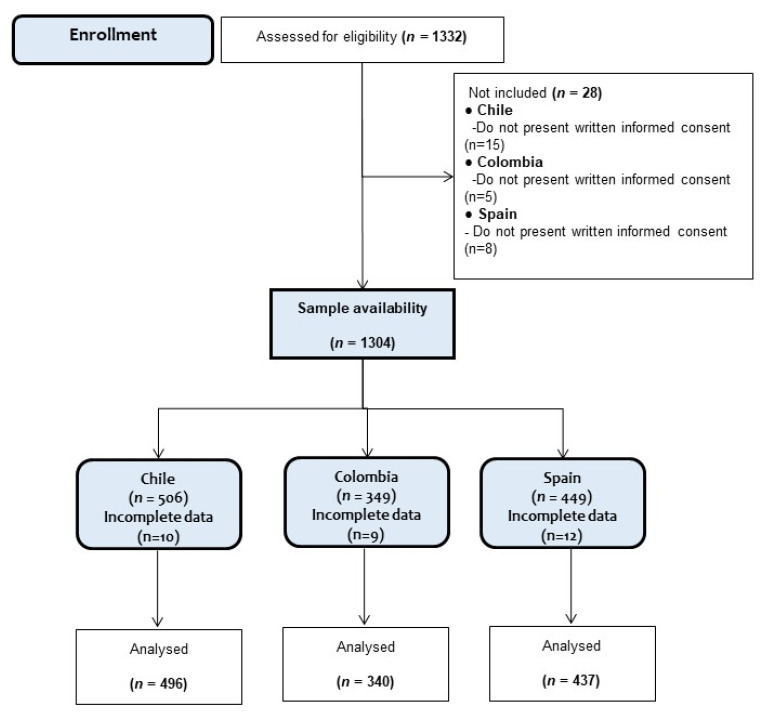
Study design.

**Figure 2 nutrients-13-02672-f002:**
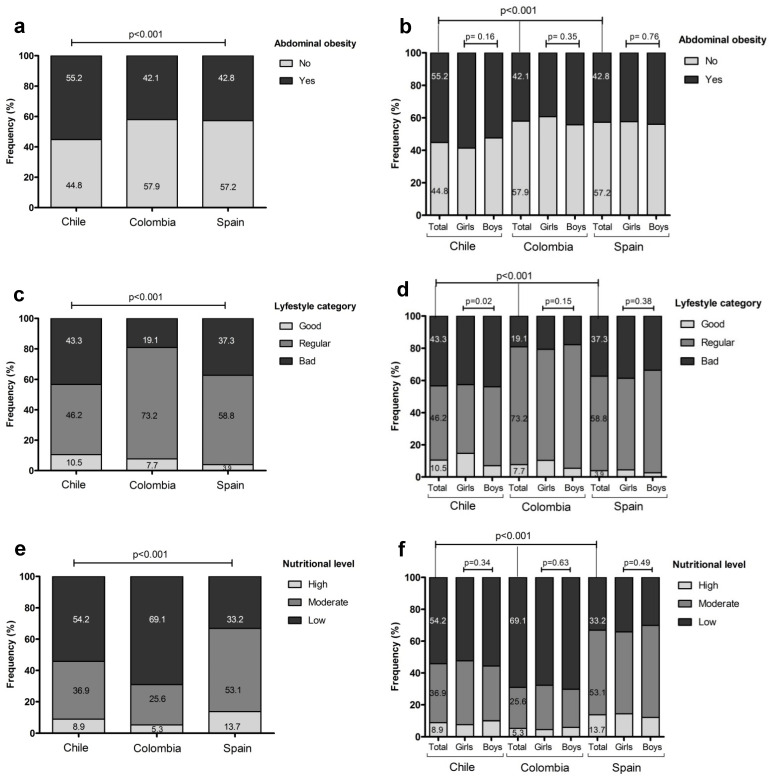
Frequency of abdominal obesity by country and sex (Panel **a** and **b**), lifestyle category (Panel **c** and **d**), and nutritional level (Panel **e** and **f**) in children according to country and sex, for Chile, Colombia, and Spain.

**Table 1 nutrients-13-02672-t001:** Anthropometric characteristics, physical fitness, and nutritional status of the children according to their country.

Parameters	Chile ^b^ (*n* = 496)	Colombia ^c^ (*n* = 340)	Spain ^a^ (*n* = 437)	*p* Value
Age (y)	5.0 (4–6)	5.0 (4–6)	4.8 (4–6)	*p* < 0.001
Sex *n* (%)				*p* < 0.001
Girls	225 (45.4)	155 (45.6)	321 (73.5)	
Boys	271 (54.6)	185 (54.4)	116 (26.5)	
Anthropometric				
Body mass (kg)	20.0 (14.6–31.0)	19.6 (15–26.8)	20 (15.4–27.3)	*p* = 0.280
Height (m)	1.13 (1.0–1.31)	1.12 (1.01–1.25)	1.11 (1.0–1.21)	*p* < 0.001
WC (cm)	59 (51–66)	56 (50–67)	55 (45–65)	*p* < 0.001
WtHR(WC/height)	0.51 (0.43–0.61)	0.50 (0.42–0.61)	0.50 (0.40–0.57)	*p* < 0.001
Fitness				
Resistance 10 × 20 (s)	76 (64–96)	85 (60–115.5)	75 (61–121)	*p* < 0.001
SLJ (cm)	84 (46–116)	82.5 (43.5–116.5)	80 (41–111)	*p* = 0.100
Speed 20 m (s)	6 (4.68–7.7)	6.05 (4.7–7.7)	5.9 (4.6–8.9)	*p* = 0.160
Handgrip strength (kg)	5.8 (2–8)	6 (2–14)	6.2 (3–10)	*p* < 0.001
Lifestyle				
ST (h/day)	4 (1–5)	2 (1–4)	3 (1–4)	*p* < 0.001
PA after school	3 (1–5)	3 (0–5)	2 (0–5)	*p* < 0.001
Lifestyle PA + ST (score)	4.5 (1–9)	5 (2–8)	4 (1–7)	*p* < 0.001
MD adherence (score)	5 (−2–9)	3 (−1–9)	6 (3–9)	*p* < 0.001

Data are presented as mean with 95% confidence interval (CI), and proportion (percentage %). *p* < 0.05 is considered statistically significant (in lower case italics). BMI = body mass index, WC = waist circumference, WtHR = waist-to-height ratio, SLJ = standing long jump test, ST = screen time, MD = Mediterranean diet, PA = physical activity. ^a^ Country with ‘high’ gross domestic product, ^b^ Country with ‘average’ gross domestic product, and ^c^ Country with ‘lower’ gross domestic product.

**Table 2 nutrients-13-02672-t002:** Parents’ sociodemographic characteristics of study sample according to country.

Parameters	Chile ^b^ (*n* = 496)	Colombia ^c^ (*n* = 340)	Spain ^a^ (*n* = 437)	*p*-Value
**Parents’ Characteristics**				
Socioeconomic background *n* (%)				*p* < 0.001
Low/medium low	207 (41.7)	128 (37.7)	48 (11.0)	
Medium	274 (55.3)	200 (58.8)	379 (86.7)	
Medium high/high	15 (3.0)	12 (3.5)	10 (2.3)	
Education level *n* (%)				*p* = 0.237
Primary	90 (18.2)	62 (18.3)	103 (23.6)	
Secondary	192 (38.7)	135 (39.7)	165 (37.8)	
University	214 (43.1)	143 (42.1)	169 (38.7)	
Marital status *n* (%)				*p* < 0.001
Single	148 (29.8)	94 (27.7)	25 (5.7)	
Married	222 (44.8)	156 (45.9)	393 (89.9)	
Divorced	78 (15.7)	61 (17.9)	14 (3.2)	
Widowed	48 (9.7)	29 (8.5)	5 (1.2)	

Data are presented as numbers and proportions (percentage). *p* < 0.05 is considered statistically significant. ^a^ Country with ‘high’ Gross Domestic Product, ^b^ Country with ‘average’ Gross Domestic Product, and ^c^ Country with ‘lower’ Gross Domestic Product.

**Table 3 nutrients-13-02672-t003:** Association of parental sociodemographic status with anthropometric, fitness, and lifestyle parameters in children according to country.

	Chile ^b^	Colombia ^c^	Spain ^a^
	β (95%CI), *p*-Value	β (95%CI), *p*-Value	β (95%CI), *p*-Value
Socioeconomic level—Low-to-middle/low			
Anthropometric parameters			
Weight (kg)	−0.31 (−1.16–0.54), *p* = 0.470	−0.06 (−0.81–0.69), *p* = 0.870	−0.01 (−1.05–1.02), *p* = 0.970
WC (cm)	−0.30 (−1.20–0.58), *p* = 0.500	0.29 (−0.83–1.41), *p* = 0.600	0.40 (−1.42–2.22), *p* = 0.660
WtHR (WC/height)	0.0003 (−0.01–0.01), *p* = 0.950	0.002 (−0.01–0.014), *p* = 0.730	0.008 (−0.007–0.02), *p* = 0.280
Fitness			
Resistance 10 × 20 (s)	−1.16 (−3.08–0.75), *p* = 0.230	1.45 (−2.08–4.9), *p* = 0.420	−2.38 (−7.87–3.1), *p* = 0.390
SJT (cm)	**−3.90 (−7.70–0.12), *p* = 0.043**	−2.43 (−7.38–2.5), *p* = 0.330	1.61 (−4.76–8.0), *p* = 0.620
Speed 20 m (s)	−0.06 (−0.23–0.11), *p* = 0.490	0.02 (−0.19–0.24), *p* = 0.810	0.02 (−0.35–0.38), *p* = 0.920
Handgrip strength (kg)	**−0.61 (−0.95–−0.28), *p* < 0.001**	−0.45 (−1.34–0.44), *p* = 0.320	0.22 (−0.37–0.81), *p* = 0.460
Lifestyle			
ST (h/day)	0.004 (−0.27–0.28), *p* = 0.970	−0.01 (−0.25–0.23), *p* = 0.930	0.15 (−0.17–0.48), *p* = 0.350
PA after school	−0.10 (−0.31–0.10), *p* = 0.320	−0.26 (−0.62–0.08), *p* = 0.130	−**0.58 (−1.05–−0.11), *p* = 0.016**
Lifestyle PA + ST (score)	−0.11 (−0.55–0.32), *p* = 0.620	−0.25 (−0.67–0.15), *p* = 0.220	**−0.74 (−1.33–−0.14), *p* = 0.015**
MD adherence (score)	0.06 (−0.63–0.76), *p* = 0.860	0.35 (−0.30–1.0), *p* = 0.290	**−0.86 (−1.45–−0.26), *p* = 0.004**
Education level—Primary or secondary			
Anthropometric parameters			
Body mass (kg)	−0.30 (−1.15–0.54), *p* = 0.470	−0.06 (−0.79–0.67), *p* = 0.870	0.27 (−0.39–0.94), *p* = 0.410
WC (cm)	−0.38 (−1.27–0.50), *p* = 0.390	−0.93 (−2.0–0.15), *p* = 0.090	−0.60 (−1.78–0.56), *p* = 0.310
WtHR (WC/height)	−0.0006 (−0.01– 0.01), *p* = 0.900	−0.008 (−0.02–0.003), *p* = 0.140	−0.003 (−0.01–0.006), *p* = 0.460
Fitness			
Resistance 10 × 20 (s)	0.78 (−1.11–2.69), *p* = 0.410	−1.40 (−4.85–2.05), *p* = 0.420	−1.84 (−5.37–1.68), *p* = 0.300
SJT (cm)	3.2 (−0.56–6.97), *p* = 0.090	**5.87 (1.09–10.6), *p* = 0.010**	0.22 (−3.83–4.28), *p* = 0.910
Speed 20 m (s)	0.15 (−0.01–0.32), *p* = 0.070	0.12 (−0.08–0.33), *p* = 0.250	0.12 (−0.35–0.10), *p* = 0.290
Handgrip strength (kg)	−0.30 (−0.64–0.03), *p* = 0.070	0.18 (−0.68–1.05), *p* = 0.670	0.12 (−0.25–0.50), *p* = 0.520
Lifestyle			
ST (h/day)	0.10 (−0.16–0.38), *p* = 0.430	0.17 (−0.06–0.41), *p* = 0.150	−0.11 (−0.33–0.09), *p* = 0.280
PA after school	−0.03 (−0.25–0.17), *p* = 0.700	0.14 (−0.19–0.49), *p* = 0.400	−0.20 (−0.51–0.10), *p* = 0.180
Lifestyle PA + ST (score)	−0.14 (−0.58–0.28), *p* = 0.500	−0.02 (−0.43–0.37), *p* = 0.880	−0.08 (−0.47–0.29), *p* = 0.650
MD adherence (score)	−0.07 (−0.76–0.61), *p* = 0.830	**−0.63(−1.27–−0.004), *p* = 0.0480**	**−0.58 (−0.96–−0.19), *p* = 0.003**
Marital status—Single or separated			
Anthropometric parameters			
Body mass (kg)	−0.49 (−1.33–0.35), *p* = 0.250	0.31 (−0.41–1.04), *p* = 0.390	0.02 (−1.05–1.1), *p* = 0.950
WC (cm)	0.08 (−0.79–0.97), *p* = 0.840	−0.55 (−1.63–0.52), *p* = 0.310	−1.84 (−3.70–0.05), *p* = 0.050
WtHR (WC/height)	−0.002 (−0.03–0.007), *p* = 0.580	−0.007 (−0.02–0.004), *p* = 0.190	**−0.02 (−0.03–−0.00), *p* = 0.030**
Fitness			
Resistance 10 × 20 (s)	**2.48 (0.60–4.36), *p* = 0.010**	**4.66 (1.27–8.05), *p* = 0.007**	−4.39 (−10.1–1.32), *p* = 0.130
SJT (cm)	−0.49 (−4.25–3.26), *p* = 0.790	1.73 (−3.0–6.52), *p* = 0.470	1.65 (−4.9–8.26), *p* = 0.620
Speed 20 m (s)	−0.12 (−0.29–0.04), *p* = 0.150	−0.09 (−0.29–0.11), *p* = 0.390	**0.46 (0.07–0.85), *p* = 0.020**
Handgrip strength (kg)	−0.12 (−0.46–0.20), *p* = 0.460	0.62 (−0.23–1.49), *p* = 0.150	0.27 (−0.35–0.9), *p* = 0.390
Lifestyle			
ST (h/day)	−0.12 (−0.39–0.15), *p* = 0.380	−0.14 (−0.38–0.09), *p* = 0.220	−0.21 (−0.55–0.13), *p* = 0.230
PA after school	0.04 (−0.16–0.25), *p* = 0.660	−0.13 (−0.47–0.20), *p* = 0.420	0.35 (−0.14–0.87), *p* = 0.160
Lifestyle PA + ST (score)	0.16 (−0.26–0.60), *p* = 0.450	0.01 (−0.39–0.41), *p* = 0.950	0.56 (−0.06–1.18), *p* = 0.070
MD adherence (score)	−0.16 (−0.85–0.52), *p* = 0.630	**−0.75 (−1.38–−0.13), *p* = 0.018**	0.06 (−0.06–0.68), *p* = 0.850

The data shown represent OR (95% CI), model adjusted by age and sex. Values of *p* < 0.05 were considered statistically significant (in bold). The data shown represent the β coefficient (95% CI), adjusted by age and sex. BMI = body mass index, WC = waist circumference, WtHR = waist-to-height ratio, SLJ = standing long jump test. ST = screen time, MD = Mediterranean diet; PA = physical activity. ^a^ Country with ‘high’ gross domestic product, ^b^ country with ‘average’ gross domestic product, and ^c^ country with ‘lower’ gross domestic product.

**Table 4 nutrients-13-02672-t004:** Association between parental characteristics and AO, low nutritional level, and bad lifestyle in children.

Country	Chile ^b^	Colombia ^c^	Spain ^a^
	OR (95% CI) *p*-Value	OR (95% CI) *p*-Value	(95% CI) *p*-Value
Abdominal obesity			
Low/medium-low	1.21 (0.84–1.74), *p* = 0.290	1.04 (0.65–1.66), *p* = 0.840	1.25 (0.68–2.29), *p* = 0.450
Primary/secondary	1.09 (0.76–1.56), *p* = 0.620	0.88 (0.55–1.59), *p* = 0.580	1.07 (0.72–1.59), *p* = 0.710
Divorced/separated	1.2 (0.71–2.04), *p* = 0.480	1.01 (0.54–1.91), *p* = 0.950	**0.21 (0.04–0.96), *p* = 0.045**
Single	1.0 (0.68–1.48), *p* = 0.980	1.01 (0.62–1.67), *p* = 0.940	1.16 (0.54–2.45), *p* = 0.690
Low nutritional level			
Low/medium low	0.73 (0.39–1.37), *p* = 0.340	1.13 (0.41–3.15), *p* = 0.800	1.83 (0.63–5.33), *p* = 0.260
Primary/secondary	1.01 (0.70–1.44), *p* = 0.950	**2.49 (1.55–3.99), *p* < 0.001**	1.32 (0.87–2.0), *p* = 0.180
Divorced/separated	1.59 (0.93–2.72), *p* = 0.080	**2.02 (1.0–4.07), *p* = 0.048**	0.51 (0.14–1.87), *p* = 0.310
Single	0.96 (0.65–1.41), *p* = 0.850	1.34 (0.80–2.23), *p* = 0.250	0.81 (0.36–1.84), *p* = 0.620
Bad lifestyle (PA + ST)			
Low/medium low	1.28 (0.70–2.34), *p* = 0.400	1.10 (0.46–2.6), *p* = 0.820	2.07 (0.27–16.0), *p* = 0.480
Primary/secondary	1.57 (0.87–2.82), *p* = 0.120	1.80 (0.79–4.1), *p* = 0.150	0.46 (0.14–1.44), *p* = 0.180
Divorced/separated	1.10 (0.47–2.57), *p* = 0.830	1.61 (0.33–7.75), *p* = 0.540	0.43 (0.05–3.62), *p* = 0.440
Single	1.17 (0.62–2.21), *p* = 0.620	0.47 (0.19–1.13), *p* = 0.090	**0.21 (0.06–0.71), *p* = 0.012**
ST (>4 h/day)			
Low/medium low	1.2 (0.70–2.0), *p* = 0.490	1.1 (0.61–1.83), *p* = 0.820	1.59 (0.65–3.89), *p* = 0.300
Primary/secondary	**1.82 (1.0–3.07), *p* = 0.020**	0.94 (0.55–1.59), *p* = 0.820	0.97 (0.58–1.60), *p* = 0.910
Divorced/separated	1.38 (0.63–3.03), *p* = 0.410	1.78 (0.76–4.14), *p* = 0.170	0.74 (0.2–2.77), *p* = 0.660
Single	1.24 (0.71–1.76), *p* = 0.440	0.77 (0.44–1.36), *p* = 0.370	0.81 (0.31–2.07), *p* = 0.660
PA after school (≤1 h/day)			
Low/medium low	1.17 (0.81–1.68), *p* = 0.390	1.43 (0.88–2.32), *p* = 0.130	2.37 (0.82–6.83), *p* = 0.100
Primary/secondary	1.00 (0.7–1.44), *p* = 0.960	0.99 (0.62–1.56), *p* = 0.960	1.19 (0.71–1.98), *p* = 0.500
Divorced/separated	0.96 (0.57–1.62), *p* = 0.880	1.26 (0.66–2.41), *p* = 0.480	2.54 (0.32–19.9), *p* = 0.370
Single	0.90 (0.61–1.33), *p* = 0.610	1.13 (0.68–1.86), *p* = 0.620	0.61 (0.25–1.51), *p* = 0.290

The data shown represent OR (95% CI), model adjusted by age and sex. Values of *p* < 0.05 were considered statistically significant (in bold). The data shown represent the β coefficient (95% CI), adjusted by age and sex. BMI = body mass index, WC = waist circumference, WtHR = waist-to-height ratio, SLJ = standing long jump test. ST = screen time, MD = Mediterranean diet; PA = physical activity. ^a^ Country with ‘high’ gross domestic product, ^b^ country with ‘average’ gross domestic product, and ^c^ country with ‘lower’ gross domestic product.

## Data Availability

Not applicable.

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
