# Peer review of "Association between the Sociodemographic Characteristics of Parents with Health-Related and Lifestyle Markers of Children in Three Different Spanish-Speaking Countries: An Inter-Continental Study at OECD Country Level"

_nutrients, 2021, doi:10.3390/nu13082672_

Round 1

Reviewer 1 Report

It was with great pleasure that I have reviewed this cross-cultural study conducted by Álvarez et al. which aimed to determine the association between the sociodemographic background of a child’s parents and the child’s lifestyle and screen time and health markers. The manuscript is very well written and structured. I have only minor suggestions for the authors .

Some directions for future investigations should be indicated at the end of the abstract as well as in the conclusions section.

The most recent updates to international food guides should be mentioned in the Introduction:

Fernandez, M. L., Raheem, D., Ramos, F., Carrascosa, C., Saraiva, A., & Raposo, A. (2021). Highlights of current dietary guidelines in five continents. International Journal of Environmental Research and Public Health, 18(6), 2814.

Serra-Majem, L., Tomaino, L., Dernini, S., Berry, E. M., Lairon, D., Ngo de la Cruz, J., ... & Trichopoulou, A. (2020). Updating the mediterranean diet pyramid towards sustainability: Focus on environmental concerns. International Journal of Environmental Research and Public Health, 17(23), 8758.

Willett, W. (2021). Mediterranean Dietary Pyramid. International Journal of Environmental Research and Public Health, 18(9), 4568.

The authors must provide a flowchart in section 2 with all the steps taken in carrying out the present study.

Author Response

It was with great pleasure that I have reviewed this cross-cultural study conducted by Álvarez et al. which aimed to determine the association between the sociodemographic background of a child’s parents and the child’s lifestyle and screen time and health markers. The manuscript is very well written and structured. I have only minor suggestions for the authors .

Response: Dear reviewer thanks you very much for your help and suggestions, we are sure that the article improved a lot.

Some directions for future investigations should be indicated at the end of the abstract as well as in the conclusions section.

Response: Dear reviewer, thanks by this comment. Following this, we have now stated as follows;

“…In conclusion, the lifestyle and health markers of children are affected by the sociodemographic background of their parents; therefore, it is essential to develop policies that decrease these gaps, so that children who are under-resourced can reach their full potential…”

The most recent updates to international food guides should be mentioned in the Introduction:

Fernandez, M. L., Raheem, D., Ramos, F., Carrascosa, C., Saraiva, A., & Raposo, A. (2021). Highlights of current dietary guidelines in five continents. International Journal of Environmental Research and Public Health, 18(6), 2814.

Serra-Majem, L., Tomaino, L., Dernini, S., Berry, E. M., Lairon, D., Ngo de la Cruz, J., ... & Trichopoulou, A. (2020). Updating the mediterranean diet pyramid towards sustainability: Focus on environmental concerns. International Journal of Environmental Research and Public Health, 17(23), 8758.

Willett, W. (2021). Mediterranean Dietary Pyramid. International Journal of Environmental Research and Public Health, 18(9), 4568.

Response: Dear reviewer, thanks by the comment. According with this, we have now included these as follows;

“…On the other hand, there is evidence that adhering to a Mediterranean diet (MD) is associated with positive effects on health [13], and MD has been also presented as part of the solution for CRF [14]. In this sense, it has been observed that MD is rich in plant-based foods and low in animal foods, that are both healthier and exert a lower impact on the environment [15]. Nowadays, for example, the new environmental dimension according to the food production represents a major cause of ecological pressure on the natural environment, and diet links worldwide human health with environmental sustainability [16], therefore, it is important that countries of different continents, take into consideration the environment and the economy to develop of MD dietary guidelines [17,18]….”

The authors must provide a flowchart in section 2 with all the steps taken in carrying out the present study.

Response: Dear reviewer, thanks by this comment. Following your suggestion, we have now created and added a new Supplementary File 1 Flow chart about the study design.

It was with great pleasure that I have reviewed this cross-cultural study conducted by Álvarez et al. which aimed to determine the association between the sociodemographic background of a child’s parents and the child’s lifestyle and screen time and health markers. The manuscript is very well written and structured. I have only minor suggestions for the authors .

Response: Dear reviewer thanks you very much for your help and suggestions, we are sure that the article improved a lot.

Some directions for future investigations should be indicated at the end of the abstract as well as in the conclusions section.

Response: Dear reviewer, thanks by this comment. Following this, we have now stated as follows;

“…In conclusion, the lifestyle and health markers of children are affected by the sociodemographic background of their parents; therefore, it is essential to develop policies that decrease these gaps, so that children who are under-resourced can reach their full potential…”

The most recent updates to international food guides should be mentioned in the Introduction:

Fernandez, M. L., Raheem, D., Ramos, F., Carrascosa, C., Saraiva, A., & Raposo, A. (2021). Highlights of current dietary guidelines in five continents. International Journal of Environmental Research and Public Health, 18(6), 2814.

Serra-Majem, L., Tomaino, L., Dernini, S., Berry, E. M., Lairon, D., Ngo de la Cruz, J., ... & Trichopoulou, A. (2020). Updating the mediterranean diet pyramid towards sustainability: Focus on environmental concerns. International Journal of Environmental Research and Public Health, 17(23), 8758.

Willett, W. (2021). Mediterranean Dietary Pyramid. International Journal of Environmental Research and Public Health, 18(9), 4568.

Response: Dear reviewer, thanks by the comment. According with this, we have now included these as follows;

“…On the other hand, there is evidence that adhering to a Mediterranean diet (MD) is associated with positive effects on health [13], and MD has been also presented as part of the solution for CRF [14]. In this sense, it has been observed that MD is rich in plant-based foods and low in animal foods, that are both healthier and exert a lower impact on the environment [15]. Nowadays, for example, the new environmental dimension according to the food production represents a major cause of ecological pressure on the natural environment, and diet links worldwide human health with environmental sustainability [16], therefore, it is important that countries of different continents, take into consideration the environment and the economy to develop of MD dietary guidelines [17,18]….”

The authors must provide a flowchart in section 2 with all the steps taken in carrying out the present study.

Response: Dear reviewer, thanks by this comment. Following your suggestion, we have now created and added a new Supplementary File 1 Flow chart about the study design.

Reviewer 2 Report

The paper addresses an important topic of children nutrition and health. For this reason, it has a valuable and socially useful content. However, there are a number of issues, especially regarding the structure of the paper that should be improved.

The abstract should not be fragmented by using headings.

The introduction is quite short and doesn’t concretely explain the basic concepts used in the paper, such as the Mediterranean diet, screen time and others. Some of these are presented in the methodology, in terms of how they were measured in the authors’ research, but it would have been necessary to clarify their general content in the introductory section. The psychological term is repeated in the first sentence of the introduction.

Considering the subject of the paper, the introduction should have mentioned the main socio-demographic, economic and cultural characteristics related to the macro-environment of the countries analyzed in the research (only a general comparison of the three countries in terms of GDP is made, without mentioning concrete values).

No hypotheses are formulated based on the literature. Research hypotheses should be based on previous studies, and to be subsequently tested in the data analysis.

Within the methodology, the research objectives are not presented in detail, nor the research method, the emphasis is only on the presentation of the characteristics of the data collection tools. Also, it is not specified when and where the research took place, but only the countries. It is not clear how the socio-economic status of the parents was established; it seems rather a subjective assessment and a general classification. In the paper, is it not specified whether certain income classes were considered or what low, medium and high socio-economic level means?

Discussions seem appropriate, making reference to the results of other studies. However, the theoretical and practical implications of the research are vaguely mentioned. These can be stated more clearly in the conclusions section.

Author Response

The paper addresses an important topic of children nutrition and health. For this reason, it has a valuable and socially useful content. However, there are a number of issues, especially regarding the structure of the paper that should be improved.

Response: Dear reviewer thanks you very much for your help and suggestions, we are sure that the article was highly improved in grammatical, English, and clarity. Please to read our last version.

The abstract should not be fragmented by using headings.

Response: Dear reviewer, thanks by this comment. We have now applied your suggestion.

The introduction is quite short and doesn’t concretely explain the basic concepts used in the paper, such as the Mediterranean diet, screen time and others. Some of these are presented in the methodology, in terms of how they were measured in the authors’ research, but it would have been necessary to clarify their general content in the introductory section. The psychological term is repeated in the first sentence of the introduction.

Response: Dear reviewer, thanks by the comment. We have now changed it and added information as follows;

“…Negative changes in lifestyle, such as a decrease in PA and an increase in screen time (ST) in children have affected their healthy psychological, physical, and social development [9]. The PA for example has reported an association with well-being in children, instead, ST has been linked negatively with psychological and physical health [10], and several cardiometabolic risk factors (CRF) [11]. Moreover, active lifestyle with high PA levels and low ST in children has commonly been studied in the health context, however, it is reasonable wide the look expecting that its promotion also involves enhancing emotional stability, cognitive skills, psychological well-being, learning process, and in turn indirectly academic achievements [2,12], therefore, the study of factors related to PA and ST in childhood has increasing attention by the scientific community.

On the other hand, there is evidence that adhering to a Mediterranean diet (MD) is associated with positive effects on health [13], and MD has been also presented as part of the solution for CRF [14]. In this sense, it has been observed that MD is rich in plant-based foods and low in animal foods, that are both healthier and exert a lower impact on the environment [15]. Nowadays, for example, the new environmental dimension according to the food production represents a major cause of ecological pressure on the natural environment, and diet links worldwide human health with environmental sustainability [16], therefore, it is important that countries of different continents, take into consideration the environment and the economy to develop of MD dietary guidelines [17,18]…..”

Considering the subject of the paper, the introduction should have mentioned the main socio-demographic, economic and cultural characteristics related to the macro-environment of the countries analyzed in the research (only a general comparison of the three countries in terms of GDP is made, without mentioning concrete values).

Response: Dear reviewer, thanks by the comment. In this line we have adde more information as follows;

“…Childhood is a relevant stage of life during which many physiological and psychological changes that are crucial for the later stages of life take place [1]. Unfortunately, the lifestyle of children today is affected when their parents have a low socioeconomic level [2]. Likewise, it is important to mention that children from schools with a low socioeconomic level had reported more barriers and fewer facilitators to obtain a healthy lifestyle with good levels of physical activity (PA) than their counterparts from schools with a high socioeconomic level [3]. A previous study has indicated that health is correlated with people’s socioeconomic status and lifestyle [4]. Concerning this, it has been reported that the socioeconomic development of the country, measured as per capita gross domestic product, is the most strongly variable associated with better health in different European countries (after controlling for individual-level characteristics) [5]. For example, Olaya et al. [6] indicated that better macro-economic indicators are related to lower numbers of children who are overweight. Additionally, the current evidence mentions that parents’ sociodemographic factors have pronounced effects on the prevalence of excess weight and obesity in children [7]. Thus, a recent study reported that socioeconomic status was strongly associated with childhood obesity early at the elementary school, where Hispanic children showed a 60% to acquire obesity than white American peers (8). On the other hand, in Latin America, important negative changes also have been described by other sociocultural factors (i.e., lower educational and socioeconomic levels from parents), and these produce a nutritional transition process [9].

Negative changes in lifestyle, such as a decrease in PA and an increase in screen time (ST) in children have affected their healthy psychological, physical, and social development [9]. The PA for example has reported an association with well-being in children, instead, ST has been linked negatively with psychological and physical health [10], and several cardiometabolic risk factors (CRF) [11]. Moreover, active lifestyle with high PA levels and low ST in children has commonly been studied in the health context, however, it is reasonable wide the look expecting that its promotion also involves enhancing emotional stability, cognitive skills, psychological well-being, learning process, and in turn indirectly academic achievements [2,12], therefore, the study of factors related to PA and ST in childhood has increasing attention by the scientific community.

On the other hand, there is evidence that adhering to a Mediterranean diet (MD) is associated with positive effects on health [13], and MD has been also presented as part of the solution for CRF [14]. In this sense, it has been observed that MD is rich in plant-based foods and low in animal foods, that are both healthier and exert a lower impact on the environment [15]. Nowadays, for example, the new environmental dimension according to the food production represents a major cause of ecological pressure on the natural environment, and diet links worldwide human health with environmental sustainability [16], therefore, it is important that countries of different continents, take into consideration the environment and the economy to develop of MD dietary guidelines [17,18].

Looking at the health markers for children, childhood obesity at the preschool stage is a global phenomenon in developed and undeveloped countries, where the nutritional transition (i.e., the change from underweight to an overweight disease due to the industrialization processes) can explain the coexistence of obesity and nutritional deficiencies; besides, positive energy balance behavior is associated with obesity [19]. Likewise, abdominal obesity (AO) is a major clinical and public health issue compared with generalized obesity in children, and due to AO, representing central obesity, this is more strongly correlated with CRF [20] and seems to have increased during the last few decades [21]. Moreover, it has been reported that AO in children increases CRF such as dyslipidemia, hypertension, and hyperglycemia [20]. Another negative effect of a bad lifestyle on health markers is a decrease in physical fitness, which is considered to be one of the most important health markers across the life course [22] because it is a powerful marker of health in early childhood and later in life [23,24]. Moreover, lower fitness has been associated with higher body fatness parameters and higher blood pressure levels in schoolchildren [25].

On the other hand, the goal of the Organization for Economic Cooperation and Development (OECD) is to shape policies that foster prosperity, equality, opportunity, and well-being for all, and its countries members work with other peers, organizations, and stakeholders worldwide to address the pressing policy challenges of our time. Concerning this, we should bear in mind that the gross domestic product (GDP) per capita is different by country, and could affect lifestyle and child development. Spain has a high GDP per capita, Chile has an average GDP per capita, and Colombia has a lower GDP per capita, and these three countries present different sociocultural characteristics. An evaluation of children’s lifestyles in countries with different economic and sociodemographic characteristics provides very relevant information for the development of policies that contribute to the integral development of children, and give a better understanding of the factors associated with children’s lifestyle. Therefore, the purpose of this study was to determine, through a cross-cultural study in Chile, Colombia, and Spain, the association between the sociodemographic backgrounds of parents (i.e., socioeconomic level, marital status, and educational level) with the lifestyle (i.e., MD, PA after school and ST) and health markers of their children…”

No hypotheses are formulated based on the literature. Research hypotheses should be based on previous studies, and to be subsequently tested in the data analysis.

Response: Dear reviewer, thanks by the comment. As we consider that you are right, and agree with this, we have no eliminated the sentence;

“…Concerning this, we should bear in mind that the gross domestic product (GDP) per capita is different by country, and could affect lifestyle and child development…”

Thus, now the last paragraph of the “Introduction” section, is as follows;

“…On the other hand, the goal of the Organization for Economic Cooperation and Development (OECD) is to shape policies that foster prosperity, equality, opportunity, and well-being for all, and its countries members work with other peers, organizations, and stakeholders worldwide to address the pressing policy challenges of our time. Spain has a high GDP per capita, Chile has an average GDP per capita, and Colombia has a lower GDP per capita, and these three countries present different socio-cultural and economic characteristics. An evaluation of children’s lifestyles in countries with different economic and sociodemographic characteristics provides very relevant information for the development of policies that contribute to the integral development of children, and give a better understanding of the factors associated with children’s lifestyle. Therefore, the purpose of this study was to determine, through a cross-cultural study in Chile, Colombia, and Spain, the association between the sociodemographic backgrounds of parents (i.e., socioeconomic level, marital status, and educational level) with the lifestyle (i.e., MD, PA after school and ST) and health markers of their children…”

Within the methodology, the research objectives are not presented in detail, nor the research method, the emphasis is only on the presentation of the characteristics of the data collection tools. Also, it is not specified when and where the research took place, but only the countries. It is not clear how the socio-economic status of the parents was established; it seems rather a subjective assessment and a general classification. In the paper, is it not specified whether certain income classes were considered or what low, medium and high socio-economic level means?

Response: Dear reviewer, thanks by the comment. Following this, and taking into account other similar, we have now improved this section adding more information, as follows;

“…2.4. Sociodemographic background of parents

An ad hoc sociodemographic questionnaire was used; information such as educational level, marital status, and socioeconomic background (based on the parents’ socioeconomic self-perception) was collected from the parents by personal interviews of a member of the school staff. This procedure were carried out into a comfortable room in each school by country. The questionnaire was added to the Krece Plus at the beginning…..”

Discussions seem appropriate, making reference to the results of other studies. However, the theoretical and practical implications of the research are vaguely mentioned. These can be stated more clearly in the conclusions section.Response: Dear reviewer, thank by the comments. We have now added information as follows;

“..In addition, considering that socio-cultural gaps can be a permanent barrier of health satus among children of different countries members of the OECD group, the reduction in the economic gaps promoted at school from the high to those children of low economic status appear as one of the most practical strategy among countries of the OECD members…”